# The Role of Electrical Polarity in Electrospinning and on the Mechanical and Structural Properties of As-Spun Fibers

**DOI:** 10.3390/ma13184169

**Published:** 2020-09-19

**Authors:** Daniel P. Ura, Joan Rosell-Llompart, Angelika Zaszczyńska, Gleb Vasilyev, Arkadiusz Gradys, Piotr K. Szewczyk, Joanna Knapczyk-Korczak, Ron Avrahami, Alena O. Šišková, Arkadii Arinstein, Paweł Sajkiewicz, Eyal Zussman, Urszula Stachewicz

**Affiliations:** 1International Centre of Electron Microscopy for Materials Science, Faculty of Metals Engineering and Industrial Computer Science, AGH University of Science and Technology, 30-059 Kraków, Poland; urad@agh.edu.pl (D.P.U.); pszew@agh.edu.pl (P.K.S.); jknapczyk@agh.edu.pl (J.K.-K.); 2Department of Chemical Engineering, Universitat Rovira i Virgili, Av. dels Països Catalans 26, 43007 Tarragona, Spain; joan.rosell@urv.cat; 3Catalan Institution for Research and Advanced Studies-ICREA, Pg. Lluís Companys 23, 08010 Barcelona, Spain; 4Laboratory of Polymers and Biomaterials, Institute of Fundamental Technological Research, Polish Academy of Sciences, 02-106 Warszawa, Poland; angelika.zaszczynska@gmail.com (A.Z.); arkadiuszgradys@gmail.com (A.G.); psajk@ippt.pan.pl (P.S.); 5NanoEngineering Group, Faculty of Mechanical Engineering, Technion–Israel Institute of Technology, 32000 Haifa, Israel; mevasil@me.technion.ac.il (G.V.); ronavra@technion.ac.il (R.A.); mearin@technion.ac.il (A.A.); meeyal@technion.ac.il (E.Z.); 6Polymer Institute of Slovak Academy of Sciences, 845 41 Bratislava, Slovakia; alena.siskova@savba.sk

**Keywords:** fibers, electrical polarity, charges, electrospinning, PMMA, mechanical properties

## Abstract

Electric field strength and polarity in electrospinning processes and their effect on process dynamics and the physical properties of as-spun fibers is studied. Using a solution of the neutral polymer such as poly(methyl methacrylate) (PMMA) we explored the electrospun jet motion issued from a Taylor cone. We focused on the straight jet section up to the incipient stage of the bending instability and on the radius of the disk of the fibers deposited on the collecting electrode. A new correlation formula using dimensionless parameters was found, characterizing the effect of the electric field on the length of the straight jet, ${\tilde{L}}_{E}\sim {\tilde{E}}^{0.55}.$ This correlation was found to be valid when the spinneret was either negatively or positively charged and the electrode grounded. The fiber deposition radius was found to be independent of the electric field strength and polarity. When the spinneret was negatively charged, ${\tilde{L}}_{E}$ was longer, the as-spun fibers were wider. The positively charged setup resulted in fibers with enhanced mechanical properties and higher crystallinity. This work demonstrates that often-overlooked electrical polarity and field strength parameters influence the dynamics of fiber electrospinning, which is crucial for designing polymer fiber properties and optimizing their collection.

## 1. Introduction

In the nanotechnology era, it is crucial to understand the relationship between the properties of materials and their structure at the macro and nano scales. Such knowledge can be applied to design well-controlled manufacturing processes, reducing the need for post-processing. It is known that the amorphous structure of polymers can be partially organized under the influence of external forces [1], providing the opportunity to tune their mechanical properties [2,3]. Electrospinning is the primary method used to produce fibers at the sub-micron level at both lab and industrial scales [4,5] including polymer composites and membranes [6,7,8]. Electrospun fibers have unique properties due to flexibility in processing [9,10] that is controlled by many parameters, including ambient conditions, polymer solution rheology and viscoelastic properties, solution conductivity, solution flow rate, nozzle-collector distance and the applied voltage [11,12]. The electrospinning setup can have various configurations including high voltage and ground connections that affects the electric field strength and shape, including charge distribution [13,14,15]. The applied electric field offers a great advantage in tuning the material properties. The fibers’ surface energy and surface potential can be modified by switching the electrical polarity of the nozzle, as was shown with polyamide 6 (PA6) [16], poly (ε-caprolactone) (PCL) [17], poly(vinylidene fluoride) (PVDF) [18] and poly(methyl methacrylate) (PMMA) [19] fibers. Importantly, the surface properties have a direct impact on wetting and cell response in in-vitro studies, as well as on triboelectric performance in energy-harvesting applications [20]. However, there is little knowledge about the effect of electrical polarity on the mechanical and structural properties of electrospun polymer fibers.

In this study, we investigate the effect of the positive or negative voltage on the mechanical performance of aligned and random poly(methyl methacrylate) (PMMA), fiber mats. We focus on the changes occurring in the electrospinning process, while positive or negative charges accumulate on the polymer solution jet [21,22]. We explore the process dynamics and concentrate on the length of the straight section of the jet and the area of the deposited fibers on the collecting electrode. The mechanical and structural properties of the resulting fibers were analyzed using differential scanning calorimetry (DSC), molecular mass measurement, material density with gas pycnometer and tensile tests. By tensile testing both aligned and random PMMA samples, we draw conclusions relating to the mechanical performance of electrospun fiber mats. The tensile strength of random fibers showed the importance of surface properties of fibers produced with positive and negative electrical polarities and the inter-fiber interactions in the mechanical performance of the fiber mat.

### Electrical Field Polarity and Strength

In a typical electrospinning process, when an electric field applied to a liquid droplet exceeds a critical value, the droplet is stretched, forming a Taylor cone and a liquid jet is ejected from the cone vertex [23,24,25]. A stable jet is characterized by an initial straight section of length $L_{E}$, stretched between the cone vertex and a fixed point along the jet, where the well-known bending instability begins. From then on, the jet follows a spiraling path in three dimensions, wherein for each loop, the circumference diameter is increased. Finally, the polymer jet is deposited on the collecting electrode, forming a disk with a radius $R_{E}$ (Figure 1a). Under certain conditions, during jet elongation, branching of the jet into smaller jets is observed [26,27]. Several models describing the electrospinning process and the behavior of the polymer jet were proposed; nonetheless, the models did not consider the effect of the electrical polarity on the electrospinning process and the relation to the spinnability of polymer-solvent systems [24,28,29,30,31,32,33,34,35].

The charge distribution and charge flow direction during electrospinning with positive or negative voltage are illustrated in Figure 1b,c. When the electrospun solution is a poorly conducting leaky dielectric fluid, net charge accumulates at the cone and jet surfaces [28,36,37]. The charge is of the same polarity as the applied voltage of the spinneret. When the spinneret is positively charged (Figure 1c), the flying jet carries excess (or net) positive charges (usually ions) on its surface, while the bulk of the jet carries cations and anions that have not had yet migrated to the surface of the jet impelled by the electrical field inside the jet. When the spinneret is negatively charged, the situation is reversed (Figure 1b). Most models of electrospinning thus consider the transport of charges by conduction, dependent on a single parameter, the electrical conductivity of the liquid phase.

The differences in system behavior depending on polarity may arise only from differences in the anion and the cations, for example, electrical mobility and/or chemical affinities for the dissolved polymeric species. Differences in electrical mobility of the ionic species may lead to changes, however slight, in the distribution of the ions near the interface [38,39,40]. On the other hand, a different chemical affinity of the cations and the anions in solution for the dissolved polymer chains, a different distribution of the polymer chains and/or of their orientations would be expected depending on the polarity used and this difference could lead to differences in the development of the jet. As surface tension is a major player in the Taylor cone and jet development (as will be justified later) and it is often dependent on the polymer surface concentration, the mentioned polarity-dependent polymer-ion affinities could influence the development of the jet by causing differences in the surface tension. In conclusion, although the effects of polarity reversal are expected to be small, they may exist for various reasons. Thus, for example, when the spinneret is negatively charged, the quantity of net charges of the flying jets could be smaller, resulting in a smaller electric current, as in fact, we have found.

The volume charge density, which describes the amount of net electric charge per unit volume after initiation of the electrospinning process, in the absence of solvent evaporation is indicated as [26]:(1)$$\rho =\frac{I}{Q},$$
where Q is the volumetric flow rate of the polymer solution, assumed constant and I is the electrical current transported along with the jet. The local volumetric charge density along the jet increases from this initial value, due to solvent evaporation, assuming no loss of charge. The current in the process is the sum of two contributions—surface convection of net charges and ohmic bulk conduction. As the jet develops and stretches, convection of surface charges becomes more significant. Eventually, surface charge convection becomes the dominant charge transport mechanism [41] and one can relate the local volume charge density ρ to the local surface charge density q as [26]:(2)$$q=\frac{\rho d}{4},$$
where d is the measured diameter of the jet at the location under consideration and a slowly tapering jet with uniform velocity profile is assumed (satisfied by the highly viscous jets encountered in electrospinning). When the solvent has not significantly evaporated from the Taylor cone and the jet up to the location where Equation (2) is used, then ρ in this equation can be predicted from Equation (1). Solvent evaporation and jet stretching and branching is known to cause a variation in the surface charge density at the jet surface [24,26,31,32,42]. However, we consider the effect of the electrical polarity and field strength on polymer jet length $L_{E}$ during electrospinning and, consequently, on the mechanical and microstructural properties of the electrospun fibers, as well as the effect of the polarity on the area of the deposited fibers.

## 2. Materials and Methods

### 2.1. Electrospinning Process

PMMA (*M*_w_= 350.000 g mol^−1^, Sigma Aldrich, Gillingham, UK) was dissolved in N,N-dimethylformamide (DMF, Sigma Aldrich, Gillingham, UK) to obtain a transparent polymer solution with a concentration of 12 wt%. The polymer was dissolved at 55 °C using the magnetic stirrer plate, set at a rotation speed of 700 rpm (IKA RCT basic, Staufen, Germany) for 2.5 h.

Electrospinning of PMMA was performed with an EC-DIG apparatus, with a climate-controlled chamber system (IME Technologies, Waalre, the Netherlands), at a temperature (T) of 25 °C and relative humidity (RH) of 40% to produce random (R) and aligned (A) fibers. Positive and negative potential of 12.00 kV, denoted as PMMA+ and PMMA−, respectively, was applied. The distance between the needle (spinneret) and the collector was 15 cm and the flow rate of the polymer solution Q was 4.0 mL·h^−1^. The inner needle diameter was 0.8 mm. Aligned PMMA fibers were collected on a drum with a 10 cm outer diameter, rotating at 2300 rpm. The fibers were deposited for 15 min on Al foil. The jet length, $L_{E}$, was measured from the tip of the cone at the needle orifice, to the point where the jet showed instability and bending began (Figure 1a). The diameter of the jet, d, was measured within 2.4 mm from the end of the needle. Both $L_{E}$ and d were determined from images taken with a DSLR camera (EOS 700D, lens EF-S 60mm f/2.8 Macro USM, Canon, Tokyo, Japan), see Appendix A. The images of the deposition areas and polymer jet length, collected from 10 samples, were analyzed using ImageJ software (J1.46r, Fiji, Madison, WI, USA). The area of the deposited fibers was imaged after 1 min of electrospinning.

### 2.2. Electrical Current

For measurement of the electrical current during electrospinning, I, a new setup was assembled. The needle was mounted and oriented vertically and the electrospinning was carried out at ambient conditions of *T* = 22 °C ± 1 °C and *RH* = 35% ± 1%. The polymer solution flow rate was set at Q = 4.0 mL·h^−1^ by a PHD 2000 infusion pump (Harvard Apparatus, Holliston, MA, USA). The high-voltage output from two power supply modules with output ranging from 0 to + and −15 kV, respectively (HV-RACK-4-250, UltraVolt, Denver, CO, USA), was connected to the needle through a 250 MΩ resistor. The needle voltage was set to 12 kV (positive or negative voltage polarity) and was measured on the needle-side of this resistor using a high-voltage probe (1 GΩ with division ratio 1900:1, TT-HVP 40, Testec, Frankfurt, Germany), whose output was connected to a digital oscilloscope (WaveJet 314, LeCroy, Chestnut Ridge, NY, USA: 1 MΩ input resistance). Another oscilloscope channel was directly connected to the collector electrode, a square (10 cm × 10 cm) brass plate, which was resting on an insulating (Teflon) block. In a typical experiment, the polarity was switched several times in the course of about 30 min. Voltage and current data were acquired for 1 s and saved at various times during each experiment, always 240 s after switching. The electrical current data was acquired when the polymer solution flowed through the system and fibers were deposited on the brass plate, whereas the current baseline was obtained for each experiment when there was no polymer solution flow. The average and standard deviations for the applied voltage were calculated from three measurements each for positively and negatively voltage. The collected current was computed from the voltage recorded on the oscilloscope using Ohm’s law:(3)$$I=\frac{V_{C}-V_{baseline}}{R_{Scope}},$$
where *I* is the electrical current on the collector, *V**_C_* is the voltage building upon the collector and measured on the oscilloscope, *V_baseline_* is the baseline voltage measured on the same channel without a flow of polymer and *R_Scope_* is the input resistance of the oscilloscope (1 MΩ = 1 mV·nA^−1^); see Appendix A. It is important to note that the current depends on the flow rate, voltage and distance between the needle and the collector plate (among other parameters) [26]. Therefore, it is critical to maintain these parameters constant (absolute value for voltage) when comparing the effect of electrical polarity on electrospun fibers.

### 2.3. Fiber Morphology and Contact Angle

Characterization of the surface and cross-sectional area of freeze fractured PMMA electrospun fibers, was performed using scanning electron microscopy (SEM, Merlin Gemini II, ZEISS, Oberkochen, Germany) and tensile tests. Fiber samples were collected on Al foil sheets placed on the collectors. After collection and 24 h drying, they were coated with a 5 nm Au layer in a rotary pumped coater (Quorum Q150RS, Quorum Technologies Ltd., Lewes, UK). The fibers were imaged under a 2.5 kV accelerating voltage, 110 pA current and a working distance of 4–9 mm. Fiber diameter ($D_{f}$ was determined from the SEM images using a plug-in tool in ImageJ (J1.46r, Fiji, Madison, WI, USA). The average diameter $\bar{D_{f}}$ and standard deviation $\sigma \left(D_{f}\right)$ were calculated from 100 randomly selected measurements of fiber diameters for each type of sample. Freeze-fractured analysis was performed for at least five fibers per sample type.

Advancing contact angles (Ɵ) on randomly PMMA electrospun membranes, deposited on glass slides, were measured using 3 liquids with different surface tension (γ)—deionized (DI) water (γ = 72.2 mJm^−2^, Spring 5UV purification system, Hydrolab, Straszyn, Poland,), glycerol (γ = 64 mJm^−2^, Pure, Sigma Aldrich, Gillingham, UK) and formamide (γ = 58.5 mJm^−2^, Pure, Sigma Aldrich, Gillingham, UK), as in previous studies [43]. The images of droplets were taken using DSLR camera (EOS 700D, lens EF-S 60mm f/2.8 Macro USM, Canon, Tokyo, Japan), after 5 s from the deposition of 3 μL droplets on membranes. Experiments were carried out at *T* = 25 °C and *RH* = 45%. The contact angles were measured for 10 different droplets deposited on fibers using drop shape analysis plug-in in ImageJ (version J1.46r, Fiji, Madison, WI, USA), with the results presented in Appendix A.

### 2.4. Crystallinity, Density and Molecular Mass of PMMA Fiber Membranes

The crystallinity of the PMMA fibers was determined using a differential scanning calorimeter (DSC, Pyris 1, Perkin Elmer, Waltham, MA, USA) operating under a nitrogen purge. Samples (ca. 10 mg) were cut from PMMA membranes and placed in standard Al pans. Scans were heated from 20 °C to 210 °C, at a rate of 10 K·min^−1^. The melting heat, $\Delta H_{f}$, measured from the area of the melting peak (Origin software 2019b, OrginLab, Northampton, MA, USA), was taken for determination of crystallinity,
(4)$$x_{c}=\frac{\Delta H_{f}}{\Delta H_{f}^{0}},$$
where $\Delta H_{f}^{0}$ is the heat of fusion for 100% crystalline PMMA, taken as 96 J·g^−1^ [44]. Average values of crystallinity and melting temperature were obtained from three measurements. DSC heating scans are shown in Appendix A and the glass transition temperatures of the PMMA power and fiber samples in Appendix A.

The density of PMMA samples was measured using a gas pycnometer (AccuPyc 1330 He, Micromeritics, Norcross, GA, USA), equipped with a 1 cm^3^ cylinder filled with a weight of 0.137 g ± 0.001 g. The average density value was calculated from 20 measurements.

Gel permeation chromatography (GPC) measurements were performed with an Agilent technology 1260 Infinity system with three column configurations (8 mm × 300 mm SDV-type columns with 5 μm beads and porosity 100.000 Å, 1000 Å 100 Å, provided by Polymer Standards Services GmbH, Mainz, Germany). The system was operated with tetrahydrofuran (THF), as eluent, at a flow rate of 1 mL·min^−1^. The concentration of the samples was 10 mg·mL^−1^. The columns were tempered at 35 °C. RI detector and PMMA calibration (also provided by Polymer Standards Services GmbH, Mainz, Germany) were used. The results are presented in Appendix A.

### 2.5. Mechanical Testing of PMMA Fibers Mat

PMMA fibers were deposited during electrospinning, on specially designed 20 mm × 8 mm paper laser-cut rectangular frames, which were later used in a tensile module equipped with a 1N cell (B.1708.A, Kammrath & Weiss, Dortmund, Germany), at *T* = 24 °C and *RH* = 50%, at strain rate 20 µm·s^−1^. Importantly, for the aligned fibers, all the samples were tested parallel to the axis of elongation. The average values of maximum stress (σ_max_), toughness (*W*), Young’s modulus (*E*_Y_), strain at maximum stress (ε_max_) and strain at failure (ε_f_) were calculated. Stress was calculated as force measured by the tensile module to initial (before elongation) cross-sectional area of the electrospun fibers mat. The Young’s modulus, *E*_Y_, was determined at the strain range of 0.1–0.5%. Sample thickness was measured by SEM; see Appendix A.

### 2.6. Statistical Analyses

The statistical analysis of the jet length, area of deposition and fiber diameter was performed using OrginPro (ver. 2020 SR1, OriginLab, Northampton, MA, USA) software, using Student’s *t*-test. For all tests, the significance was set at *p* < 0.05. The data are expressed as the arithmetic average ± standard deviation (SD). The average values of electrical current, crystallinity, density, molecular weight and mechanical properties were calculated from 5 measurements.

## 3. Results and Discussion

### 3.1. Electrospinning

A focus of the experiments was the straight jet, $L_{E}$, that emanates from the Taylor cone until a point along the jet where the bending instability began. Another was the radius of the collection area, $R_{E}$, where the electrospun fibers were deposited. Electrospinning was carried with positive and negative voltage, with a fixed electric field strength E = 80 kV·m^−1^ and flow rate Q = 4.0 mL·h^−1^. For positive voltage (PMMA+), the length of the straight section of the jet after emerging from the needle (spinneret) was $L_{E}$ = 2.9 cm ± 0.1 cm and the radius of the disk of fibers deposited on the collector was $R_{E}$ = 28.7 cm ± 6.3 cm. For negative voltage (PMMA-), the straight jet length was longer, reaching $L_{E}$ = 3.2 cm ± 0.2 cm, while the disk of deposited fibers was smaller, with $R_{E}$ = 24.2 cm ± 8.7 cm (Figure 2a–d). The polymer jet length in the straight section is dependent on the polymer-solvent parameters, including viscoelasticity, electrical conductivity, dielectric constant and surface tension [23,24]. By using DMF to dissolve PMMA, which has a relatively low vapor pressure, with a dielectric constant of ε(DMF) = 36.7 and electrical conductivity of K(DMF) = 6 × 10^−6^ Sm^−1^ [45,46], electrical charge dissipation was rather low, allowing for storage of a higher amount of electrical energy in the liquid. The high dielectric constant of DMF encourages the ionization of species in solution, thus raising the electrical conductivity and therefore the availability of charge resulting in larger volume charge density compared to other solvents under equal situations. Also, DMF often leads to liquid-liquid phase separation during electrospinning and to interior pore formation before fiber solidification [47].

In order to link the operating parameters to the jet characteristics, we adopted the dimensionless parameters presented by Sahay et al. [23]. The dimensionless length of the jet ${\tilde{L}}_{E}$, the dimensionless radius of the deposition area ${\tilde{R}}_{E}$ and the dimensionless electrical field strength $\tilde{E}$ are:(5)$${\tilde{L}}_{E}\equiv L_{E}3\eta K/\left(\gamma \varepsilon \varepsilon_{0}\right)$$
(6)$${\tilde{R}}_{E}\equiv R_{E}3\eta K/\left(\gamma \varepsilon \varepsilon_{0}\right)$$
(7)$$\tilde{E}\equiv E\varepsilon \varepsilon_{0}/{\left(3\eta K\right)}^{\frac{1}{2}},$$
where E is the electrical field strength (dependent on the applied voltage) and the polymer solution parameters are—η viscosity, $\varepsilon \varepsilon_{0}$ dielectric permittivity (equal to the product of the dielectric constant ε, also called relative permittivity and the vacuum permittivity, $\varepsilon_{0}$ = 8.854 pF·m^−1^, which is missing in the Sahay et al.’s expressions), K electrical conductivity and γ surface tension. The values of the solution parameters are presented in Appendix A and taken from reference [48].

The electrospinning process was carried out in 12 kV of applied voltage for positive and negative electrical polarity, while maintaining all the other parameters the same. The relation between ${\tilde{L}}_{E}$ and ${\tilde{R}}_{E}$ versus $\tilde{E}$ for a positive and negative electrical polarity are presented in Figure 2e,f. By increasing the electrical field strength, through the increase of the applied voltage, the value of ${\tilde{L}}_{E}$ increased for both the positive and the negative voltage [23,24]. As expected, the length of the straight jet, ${\tilde{L}}_{E}$ was longer when the spinneret was negatively charged, for all the measured points, consistent with a smaller net charge density and therefore a delayed inception of the bending instability. The effect of the non-dimensionless electric field strength, $\tilde{E}$, on the non-dimensional radius, ${\tilde{R}}_{E}$, was negligible, although the dispersion of the data was rather large.

The data values for both the negative and the positive voltage polarities, in a log-log scale (Figure 2e), were found to collapse on a straight line, with the best-fit line having a linear slope for the correlation between the quantities defined in Equations (5) and (7). Thus, the correlation between the dimensionless length of the straight section of the electrospun jet and the dimensionless electric field is
(8)$${\tilde{L}}_{E}\;\sim \;{\tilde{E}}^{0.55}.$$

This correlation formula is valid for the analyzed flow rate of Q = 4.0 mL·h^−1^. When considering typical parameters used in the electrospinning process, in a capillary-dominated regime [30], the jet shape near the spinneret is characterized at most by one parameter - the electrical Bond number, $Bo_{E}\sim \left(a_{0}\;\varepsilon_{0}\;E^{2}\right)/\gamma$, (*a*_0_ ~ 1 mm being the inner needle radius), which determines the relative importance of electrical and capillary stresses. In the studied system $Bo_{E}>1$, hence, electrical stresses play a significant role in shaping the Taylor cone. The relative importance of viscous relative to surface-tension stresses, as measured by the capillary number $Ca\sim \eta \;Q/\left(a_{0}^{2}\;\gamma \right)$, is small for our system (Ca~0.047). However, during jet initiation and thinning, viscous stresses grow rapidly in importance, as the characteristic length is no longer $a_{0}$ but the jet width.

The electrical currents measured on the collector during electrospinning at positive and negative voltage, were 90.3 nA ± 1.4 nA and - 86.8 nA ± 0.5 nA, respectively (Appendix A). The small but repeatable differences in these values indicate that the volume charge density ρ (I/Q) changed with the electrical polarity. Based on Equations (1) and (2), the volume charge density ρ and surface charge density q for PMMA+ were 81.3 C·m^−3^ ± 1 C·m^−3^ and 5.28 × 10^−4^ C·m^−2^ ± 2 × 10^−5^ C·m^−2^, respectively and for PMMA− were - 78.1 C·m^−3^ ± 1 C·m^−3^ and - 5.84 × 10^−4^ C·m^−2^ ± 2 × 10^−5^ C·m^−2^, respectively. The values of q are characteristic of the beginning of the straight jet section, computed at 2.4 mm from the nozzle exit, where for PMMA+ the jet diameter was 26 µm and for PMMA− was 30 µm. Larger volume charge densities have been reported to anti-correlate with average fiber diameter [49] and this is consistent with our findings, where, at a constant flow rate, PMMA+ had both a higher absolute current value (|I|) and smaller fiber diameter ($\bar{D_{f}}$) than PMMA−. The changes in charge density reflect differences in the dynamics of the polymer jet formation (at a constant volumetric flow rate) [29]. For example, increased volume charge density (e.g., PMMA+) should result in a greater initial acceleration of the polymer jet (per unit electrical field strength) against the decelerating forces (elastic tension, viscous stresses and surface tension), resulting in a greater reduction in jet length, as was observed for PMMA+. As shown in Figure 2, electrospinning with positive voltage is characterized by a shorter $L_{E}$ and larger $R_{E}$. Previous works [39,50] showed that $L_{E}$ can be extended by adding salts to a polymer solution or by increasing polymer concentration in the polymer solution, resulting in a change in the critical value of the concentration of ions and charges on the polymer jet surface. The additional fitting to indicate the differences is also presented in Appendix A. Supaphol et al. [51] achieved a different shape and a larger deposition area of randomly oriented PA6 fibers for positively charged spinneret. Also, a higher fiber deposition rate, which correlated with the deposition area, was achieved by Stanger et al. [32], when a positive voltage was applied at the spinneret. In our case, positively charged spinneret (PMMA+) led to higher accumulation of charges, which affected $L_{E}$ and $R_{E}$. Therefore, the similar trends for both polarities is observed in relation of electric filed to $L_{E}$ and $R_{E}$. In summary, electrical polarity is an important parameter that enables changes in volume charge density, affecting the electrical stresses responsible for polymer jet stretching in the electric field.

### 3.2. Fibers Morphology and Microstructure

Figure 3 presents SEM micrographs of PMMA fibers and histograms of the distribution of fiber diameter. In all cases, the fibers were continuous and their surfaces were smooth. The average diameter of electrospun fibers, $\bar{D_{f}}$, in the random and the aligned fiber arrangements were 1.53 µm ± 0.20 µm and 1.55 µm ± 0.20 µm for PMMA+ and 1.54 µm ± 0.18 µm and 1.50 µm ± 0.21 µm for PMMA−, respectively. For both fiber configurations, $\bar{D_{f}}$ was smaller for the positive polarity. A Student t-test run on these data suggests that the differences in averages between polarities were not statistically significant. Size distribution and fitting distribution lines in a Gaussian model, can be found in Appendix A. The present findings were consistent with our previous measurements of random PMMA fibers [19], although the effect was more pronounced in the current setup. Interestingly, the size average ratios for the two polarities were nearly identical for the aligned and random configurations, about 7% in both cases. Small effects of electrical polarity on the fiber diameter have been previously reported with other polymer-solvent systems. In our previous research [17,18], negative polarity yielded thicker PVDF and PCL/chitosan blend fibers [52]. A negligible effect of polarity on fiber diameter has been reported for PVDF (less than 1%) [18] and for PCL [17]. Bhattacharjee et al. [53] presented results showing that the diameter of electrospun PMMA fibers was lowered with a significant increase in volume charge density. In our case, we can assume that the change in volume charge density was too small to cause a significant difference in fiber diameter. There is little information in the literature on this effect; finite correlations of fiber diameter change with polarity change have been recorded, with its sign varying from system to system. Additionally, in our case, PMMA− have a higher non-polar content at the surface, which causes a higher contact angle for three type liquids for fiber membranes, as indicated in Appendix A. Changing voltage polarity in electrospinning allowed controlling the molecular orientation of functional groups in PMMA polymer chains, which was showed previously with the XPS analysis [19].

In Figure 4, representative cross-sections of freeze-fractured electrospun PMMA fibers SEM micrographs are presented. In random and aligned PMMA+ and PMMA− fibers, a clean fracture without any cracks and necking effects is observed, similar to other investigations [54,55,56,57,58,59]. There are several studies showing that the change in the morphology, internal structure and fracture behavior of the fibers can be achieved by using different solvents [59] or PMMA blends [60,61]. In our case, non-fibrillar or fibrillar structure for PMMA was obtained by applying different voltage polarities. The cross-section of PMMA− fibers shows microfibrillar structures sticking out from the fractured fiber interior in opposition to PMMA+. This type of structure is associated with the manufacturing method, which here is related to variation in charge density in electrospinning caused by the application of positively or negatively voltage.

### 3.3. Fiber Crystallinity and Density

DSC measurements for random and aligned PMMA+ and PMMA− fibers indicated lower crystallinity levels (1–3%) for both types, compared to as-received powder (18%). The crystallinity of PMMA+ fibers, particularly those with random assembly, was higher (3%) compared to that of PMMA− samples (1%), what is significant in amorphous materials. The DSC profiles and glass transition temperature of PMMA and the crystallinity values are shown in Appendix A. For all samples, *T*_g_ was near 125 °C and was accompanied by a thermal effect related to stress relaxation, reflecting the highly non-equilibrium state of the PMMA in the fibers. Additionally, the endothermic peak, centered at ~75 °C, originated from either an overstressed state of the polymer or the presence of water absorbed during sample storage. Relatively high crystallinity of neat PMMA powder is likely the results of insufficient purification of the polymer after synthesis. The higher crystallinity for fibers produced at positive voltage, is most probably related to higher molecular orientation resulting from the shorter straight jet section and consequently more extended section with deformations, as reported above. The most commercial grades of PMMA are atactic, with randomly alternating substituents along the molecular chain. However, PMMA can contain fractions that are isotactic, syndiotactic and atactic, leading to differences in both crystallinity and molecular mass [62], as presented in Appendix A. The obtained *T*_g_ suggests the presence of a syndiotactic PMMA fraction for which the crystallization process is slow, leading to very low degrees of crystallinity. However, it is accompanied by a spherulitic structure, with the semicrystalline regions [63], characterized by self-generated fields of lamellas or fibrils in the different regions of polymers [64], affecting the mechanical properties of polymers [65].

The density measurements showed a difference between PMMA samples (see Table 1). In the case of aligned fibers, the density of PMMA− was slightly higher than that of PMMA+, which correlated with the difference in DSC crystallinity. As there is no internal porosity in our fibers (see Figure 4), the density should be proportional primarily to crystallinity. Assuming that the theoretical density of the amorphous phase in PMMA is 1.17 g·cm^−3^, which is similar for our PMMA powder ± 0.02 and the crystalline phase is 1.26 g·cm^−3^ [66,67], it is evident from the density data presented in Table 1, that the crystallinity estimated from density would be much higher than that estimated from melting heat. We propose that the relatively high density was due to denser molecular arrangements within the amorphous phase as an effect of the drawing. This denser molecular packing (orientation) within the amorphous phase can shed light on the unexpected observation that DSC crystallinity in the fibers collected on the rotating drum was not higher compared to that of randomly collected fibers. It cannot be ruled-out that the majority of the molecular reorganization in PMMA under external electric field, occurs within the amorphous phase, without incurring substantial changes in crystallinity.

### 3.4. Mechanical Properties of Fibers

The stress-strain curves of the electrospun aligned PMMA+ and PMMA− fibers and representative SEM images of fibers after specimen rupture, are presented in Figure 5. The average maximum tensile stress and toughness of aligned PMMA− were 186.8 kPa and 433.1 kJ·m^−3^, respectively (see Table 1). The average strain at maximum stress was 1.8% for PMMA− and 3.2% for PMMA+. All the properties derived from the tensile tests were higher for aligned PMMA+ than for aligned PMMA−. The Young’s modulus of aligned PMMA+ fibers reached 141.6 kPa, while for PMMA− it was 106.7 kPa. This enhanced mechanical performance with higher Young’s modulus of aligned PMMA+ can be explained by its more uniform non-fibrillar interior structure and higher crystallinity. Maximum stress for aligned fibers was consistently higher than those for corresponding random fibers. The mechanical properties of aligned PMMA fibers were affected not only by the electrical polarity but also by the rotating drum, which aligned and stretched fibers and polymer chains of individual fibers [68,69,70]. Notably, the mechanical performance of electrospun membranes depends not only on individual fibers but also on the adhesion forces between them [71]. The random orientation of fibers causes stress delocalization in the fiber mat and enhancement of mechanical performance due to the interactions between the fibers [72]. Yet, random PMMA+ fibers were characterized by lower tensile strength than random PMMA− as shown in Figure 6. The maximum tensile stress and toughness for a randomly oriented network of fibers was 129.4 kPa and 811.0 kJ·m^−3^, respectively, for PMMA+ and increased by almost 73%, reaching 228.5 kPa and 1402.0 kJ·m^−3^, respectively, for PMMA−. *ε*_max_ and *ε*_f_ were very similar for both samples, ~3% at maximum stress and 38% at failure. Table 1 summarizes the average values of the mechanical properties, which are in line with the previously reported mechanical measures of electrospun PMMA fibers [58,73,74]. Additionally, SEM images taken of random PMMA fibers after mechanical testing indicated a rigid fracture in the PMMA− sample (Figure 6b,d). While PMMA is known for its brittleness at room temperature [75], we observed more fragmentation of fibers in the negative polarity fiber mats (PMMA−). Similar rigid PMMA fibers were observed by Greenfeld and Zussman [76], immediately after electrospinning. It was explained as the result of a loss of entanglements of polymer chains during electrospinning, caused by the fiber stretching and relaxation which result from the electrostatic repulsion between charges along the polymer jet. Khanlou et al. [77] also showed the fragmentation of PMMA fibers achieved by increasing applied voltage during electrospinning, which was mostly a consequence of changes in electrical current.

## 4. Conclusions

In summary, experimental study of the electrospinning process of a PMMA provided insights into the mechanism underlying the electrical polarity-induced change in the process dynamic, mechanical properties and microstructure of the as-spun fibers. Electrical polarity is an important parameter that can control charge density, which affects the electrostatic repulsion forces that stretch polymer jets in electric fields, as measured by the jet length, fiber deposition radius and in-process current. All of these reflect differences in the dynamics of the polymer jet formation and internal structure of produced PMMA fibers, as deduced from density, DSC crystallinity measurements and tensile testing. The higher crystallinity of fibers produced at positive voltage is related to higher molecular orientation due to larger deformation of the jet, which is a function of higher charge density, shorter straight jet length and more extended section with instabilities. Subsequently, the radius of the fiber deposition area is larger. Therefore, for aligned PMMA fibers, the mechanical properties were enhanced when samples were produced with positive voltage but for the random fibers, the opposite was observed. The maximum tensile stress for random PMMA− was higher than that for PMMA+, showing that interactions between individual fibers play a crucial role in the mechanical performance of electrospun membranes. Taken together, PMMA fiber electrospinning allows for tailoring of mechanical properties by modifying electrical polarity, in a single-step manufacturing approach. Moreover, we clearly present the importance and usefulness of the often-neglected electrical polarity parameter in electrospinning, one of the most common methods to produce nano- and micro-sized polymer fibers.

## Figures and Tables

**Figure 1 materials-13-04169-f001:**
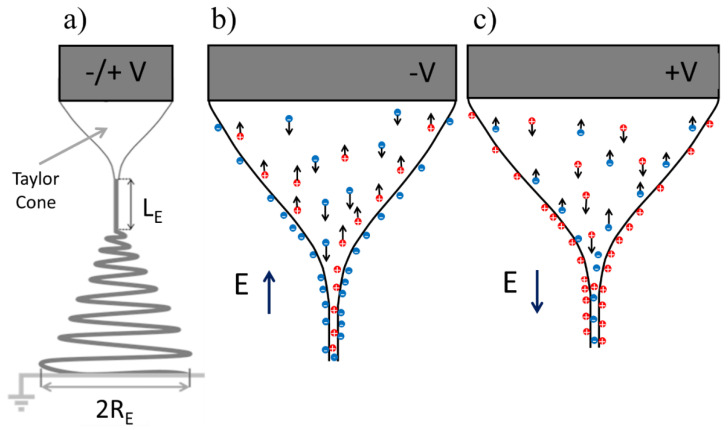
(**a**) A schematic model of the liquid jet during the electrospinning process showing Taylor cone with a straight jet section of length $L_{E}$, followed by a bending instability region, where $R_{E}$ is the radius of the deposition area. (**b**,**c**) A schematic of the Taylor cone and jet initiation, showing the charge distribution and ion motion during electrospinning as a function of positive or negative spinneret voltage, where *V* is the potential, *E* is the electrical field and red and blue dots represent clouds of positive and negative ions in solution. Black arrows indicate the electrophoretic velocity vectors of the ions.

**Figure 2 materials-13-04169-f002:**
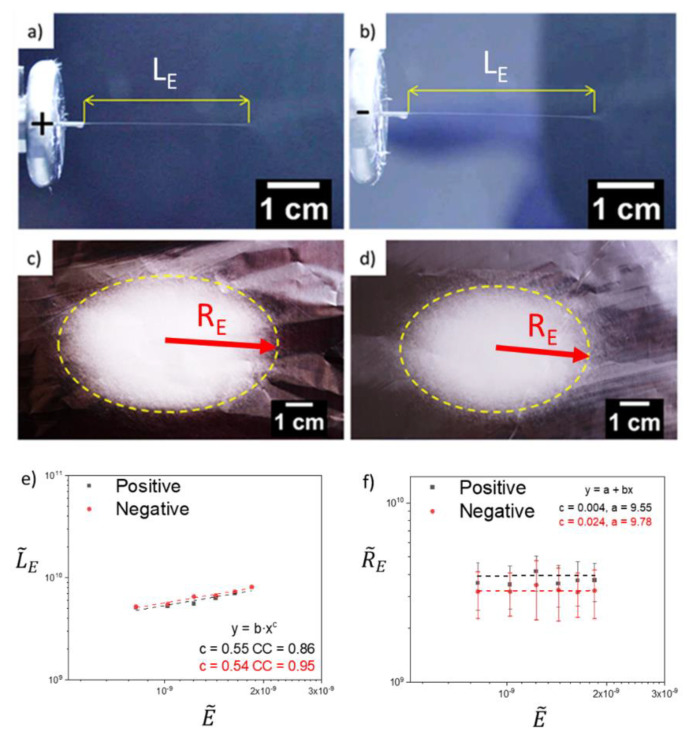
Images of the electrospinning polymer jet length before the bending instability begins and the area of deposited random fibers on Al foil for poly(methyl methacrylate) (PMMA)+ (**a**,**c**) and PMMA- (**b**,**d**), respectively. (**e**,**f**) The dimensionless length, ${\tilde{L}}_{E}$ of the jet’s straight section and the dimensionless radius ${\tilde{R}}_{E}$ of the deposition area versus the dimensionless electrical field, $\tilde{E}$ on a log-log scale. cc- adjusted r-squared goodness-of-fit.

**Figure 3 materials-13-04169-f003:**
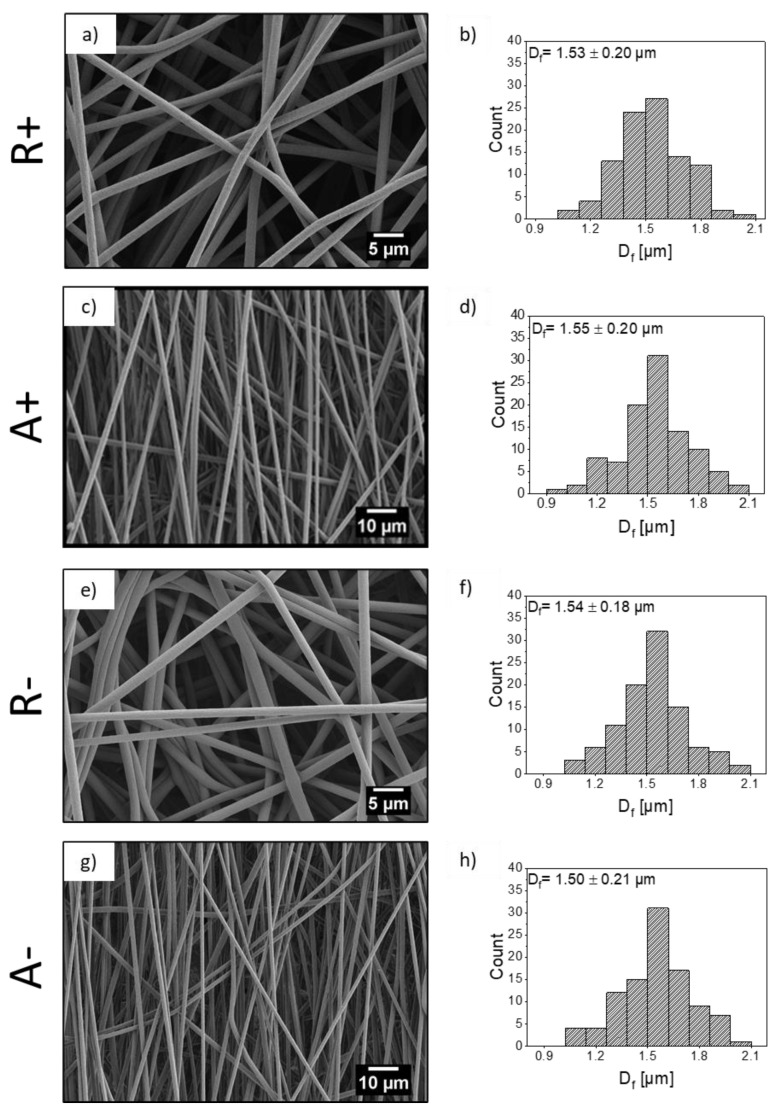
Scanning electron microscopy (SEM) micrographs of PMMA+ random, R and aligned, A fiber mats (**a**,**c**) with histograms showing the fiber diameter distribution and (**b**,**d**) PMMA− random, R and aligned, A fiber mats (**e**,**g**) with histograms showing the fiber diameter distribution (**f**,**h**).

**Figure 4 materials-13-04169-f004:**
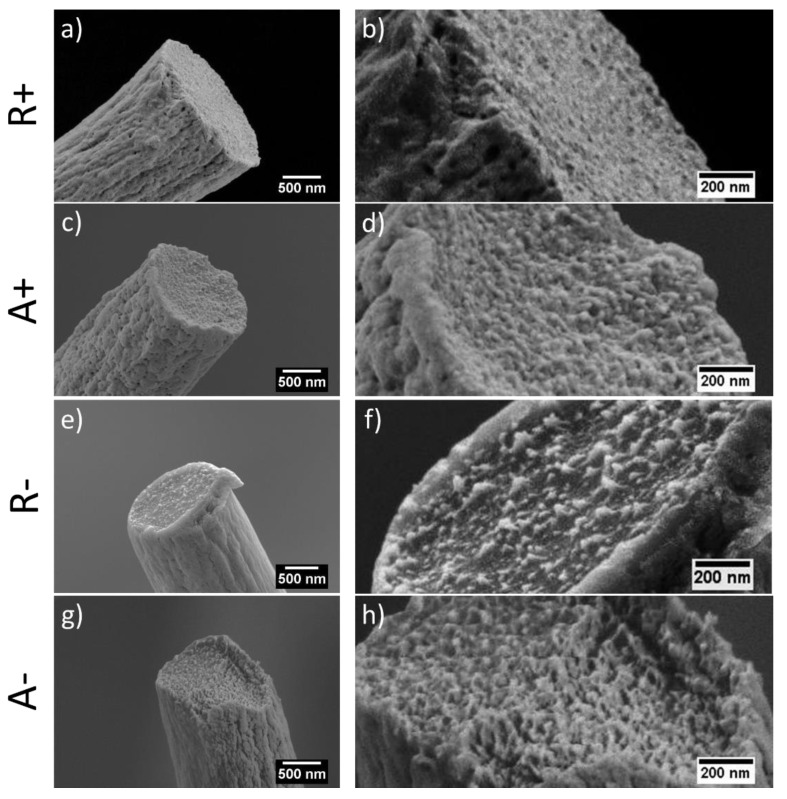
High-resolution scanning electron microscopy (SEM) micrographs of cross-sections of PMMA electrospun fibers produced with a positively (+) (**a**–**d**) or negatively (−) (**e**–**h**) charged spinneret. Fibers were deposited randomly (R) and aligned (A) on the collector.

**Figure 5 materials-13-04169-f005:**
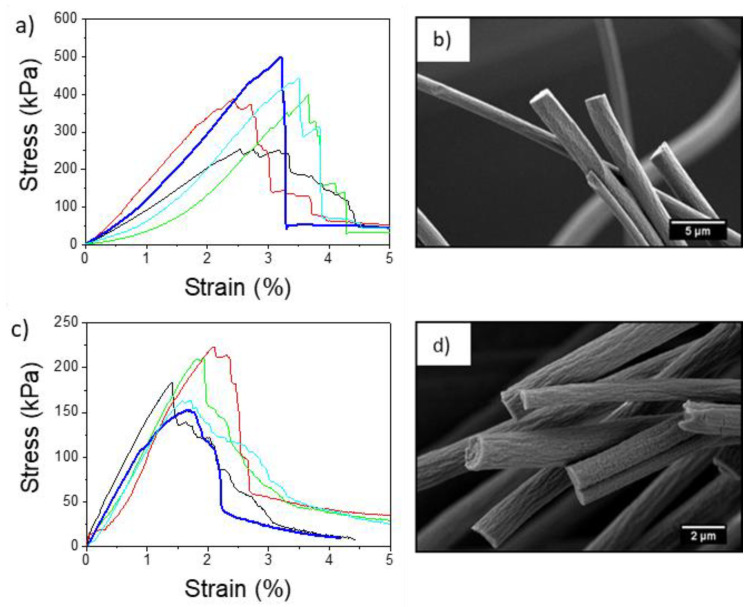
Stress-strain curve and scanning electron microscopy (SEM) images obtained after tensile testing of aligned PMMA fiber mats produced with a positively (**a**,**b**) versus negatively (**c**,**d**) charged spinneret.

**Figure 6 materials-13-04169-f006:**
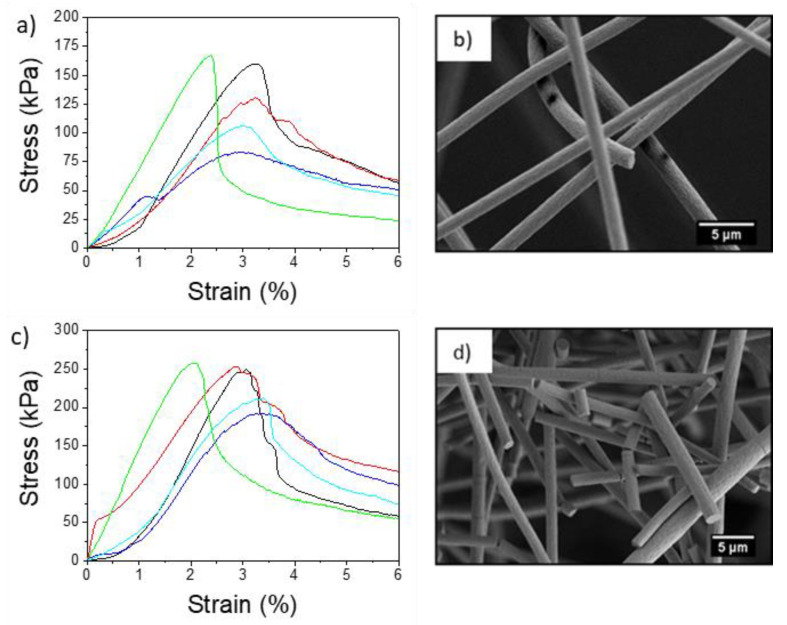
Stress-strain curves and scanning electron microscopy (SEM) images obtained after tensile testing of random PMMA fiber mats produced with a positively (**a**,**b**) versus negatively (**c**,**d**) charged spinneret.

**Table 1 materials-13-04169-t001:** Density and mechanical properties of PMMA membranes with *σ*_max_ tensile strength, *E_Y_* Young’s modulus, *W* toughness, *ε*_max_ strain at max strength, *ε*_f_ at failure. All the presented values are the average ± standard deviation. ^a,b,c,d^ indicates statistical significance between each group.

PMMA Fibers	Density [g·cm^−3^]	σ_max_ [kPa]	*ε*_max_ [%]	*ε*_f_ [%]	*E*_Y_ [kPa]	*W* [kJ∙m^−3^]
A+	1.24 ± 0.01 ^c^	446.4 ± 47.5 ^a^	3.2 ± 0.4 ^a^	17.7 ± 2.4 ^b^	141.6 ± 20.1 ^a^	1084.1 ± 188.3 ^b^
A−	1.25 ± 0.01 ^b^	186.8 ± 27.2 ^c^	1.8 ± 0.2 ^b^	9.5 ± 4.6 ^b^	106.7 ± 13.6 ^b^	433.1 ± 130.4 ^d^
R+	1.25 ± 0.01 ^a^	129.4 ± 31.7 ^d^	3.0 ± 0.3 ^a^	38.1 ± 2.6 ^a^	–	811.0 ± 162.2 ^c^
R−	1.24 ± 0.01 ^c^	228.5 ± 24.3 ^b^	3.2 ± 0.7 ^a^	37.8 ± 5.5 ^a^	–	1402.0 ± 283.3 ^a^

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
