# Peer review of "The Role of Electrical Polarity in Electrospinning and on the Mechanical and Structural Properties of As-Spun Fibers"

_materials, 2020, doi:10.3390/ma13184169_

Round 1

Reviewer 1 Report

The manuscript: The role of electrical polarity in electrospinning and on the mechanical and structural properties of as-spun fibers presents a systematic study for the preparation of PMMA fibers. The authors claim the essential effect of the spinneret charge, pointing out the role of the polarity and field strength.

The paper is well developed and easy to read. However there is a few suggestions that should be addressed.

1) materials and methods: please, indicate the number of replicates in each experiment.

2) Results and Discussion: the statistic between the different groups in table 1 is not showed, please, indicate the significance between groups.

The study of the crystallinity is well developed and complete and I suggest to include some XPS data of the films in order to explore the influence of the chemical composition in the properties. Is it possible to perform contact angle measures?

Author Response

Response to Reviewer 1:

Q1: Materials and methods: please, indicate the number of replicates in each experiment.

A1: The numbers of tests were 5. The following sentence was added for clarification see page 6 line 23h: 

'The average values of electrical current, crystallinity, density, molecular weight, and mechanical properties were calculated from 5 measurements.”

Q2: Results and Discussion: the statistic between the different groups in table 1 is not showed, please, indicate the significance between groups.

A2: Statistical significance between each group was added to the manuscript, see Table 1, page 11.

Q3: The study of the crystallinity is well developed and complete and I suggest to include some XPS data of the films in order to explore the influence of the chemical composition in the properties.

A3: XPS tests and the differences in the chemical composition of the surface of the PMMA fibers and the film have already been investigated see reference [19].

See text on page 9 line 353:

“Changing voltage polarity in electrospinning allowed controlling the molecular orientation of functional groups in PMMA polymer chains, which was showed previously with the XPS analysis [19].”

Q4: Is it possible to perform contact angle measures?

A4:  The contact angle values have been added to the Supporting Information, see Figure 1S and the rsults are discussed in the manuscript on page 9 line 351 as follows:

“Additionally, in our case, PMMA- have a higher non-polar content at the surface, which causes a higher contact angle for three type liquids for fiber membranes, as indicated in Fig. S1. Changing voltage polarity in electrospinning allowed controlling the molecular orientation of functional groups in PMMA polymer chains, which was showed previously [19].”

Also, the Method section is updated see page 5 line 197, as follows:

“Advancing contact angles (Ɵ) on randomly PMMA electrospun membranes, deposited on glass slides, were measured using 3 liquids with different surface tension (γ): deionized (DI) water (γ=72.2 mJm-2, Spring 5UV purification system - Hydrolab, Poland,), glycerol (γ=64 mJm-2, Pure, Sigma Aldrich, UK) and formamide (γ=58.5 mJm-2, Pure, Sigma Aldrich, UK), as in previous studies [37]. The images of droplets were taken using DSLR camera (EOS 700D, lens EF-S 60mm f/2.8 Macro USM, Canon, Japan), after 5s from the deposition of 3 μL droplets on membranes. Experiments were carried out at T=25°C and H=45%. The contact angles were measured for 10 different droplets deposited on fibers using drop shape analysis plug-in in ImageJ (version J1.46r, Fiji, USA).”

Reviewer 2 Report

The length of jet and electric current depend on a number of parameters. This is described in a number of papers. The authors neglect most of the parameters. Therefore the data presented do not bring any new relevant information. The discussion of results is not sufficient, should be deeper and take into account the state of knowledge in the field.

Author Response

This is described in a number of papers. The authors neglect most of the parameters. Therefore the data presented do not bring any new relevant information. The discussion of results is not sufficient, should be deeper and take into account the state of knowledge in the field.

Answer to general comment:

We thank for the comment and we agree that jet and electric current are governed by many parameters however in this study we carried out a thorough investigation on only one parameter while the rest remained constant. Such an approach allowed us to draw conclusions that agree with the current state of knowledge and expands it in a significant way as the voltage polarity parameter is crucial.

Reviewer 3 Report

Materials-904586: The role of electrical polarity in electrospinning and on the mechanical and structural properties of as-spun fibers.

In this manuscript, the authors attempt to study the effect of the electric field polarity to explain some differences in the structural characteristics and mechanical properties of the randomly collected and oriented fibers obtained by the electrospinning technique. Although the theoretical aspects of this technique are known, in most cases its application is carried out experimentally (based on trial-error) to optimize the operational parameters. In this way, the current manuscript is interesting because it proposes the re-analysis of some theoretical foundations of the technique to explain the results obtained.

Comments:

1) Line 62-63. "In this study, we verify the effect of the positive or negative voltage on the mechanical performance of aligned and random poly (methyl methacrylate) (PMMA), fiber mats." Verification assumes prior knowledge; therefore, the authors are aware that the polarity of the electric field influences the mechanical properties of the PMMA fibers matrix. However, these antecedents are not indicated in the introduction, where only changes in the energy and potential of the surface of the fibers are indicated.

2) Line 82-83. “Several models describing the electrospinning process, and the behavior of the polymer jet were proposed; nonetheless, the effects of electrical polarity were neglected [24,27–32]. ”. In these models the polarity is not forgotten, it is implicit, and the models apply for both polarities. Operationally, during the electrospinning process, it is one of the first operational parameters to be set. In most cases, the polymer dissolution with a solvent or solvent mixture only produces fibers in a certain polarity. There are few exceptional cases, polymer-solvent(s) systems that produce fibers in both polarities.

3) In this sense, (comes from comment 2) the PMMA-DMF system seems suitable for studying the effect of the polarity of the electric field on the properties of fibers. However, the new correlation function ?̃? ~ ?̃^0.55 should be demonstrated for PMMA and other solvents and / or for different polymer-solvent(s) systems, before being proposed as a generalization.

4) The new correlation function is based on the LE and RE values, taken from 10 samples. Lines 241-245: “For positive voltage (PMMA +), the length of the straight section of the jet after emerging from the needle (spinneret) was ?? = 2.9 ± 0.1 cm, and the radius of the disk of fibers deposited on the collector was ?? = 28.7 ± 6.3 cm. For negative voltage (PMMA-), the straight jet length was longer, reaching ?? = 3.2 ± 0.2 cm, while the disk of deposited fibers was smaller, with ?? = 24.2 ± 8.7 cm (Fig. 2 a-d). ”. The values ​​between the two polarities appear not to be statistically different and correspond to the same distribution. In this sense, it is understood that the new correlation function is independent of polarity. Thus, in Figure 2, the authors show extreme cases of LE and RE, but they could also show similar cases. If the authors perform a statistical analysis of the LE vs E and RE vs E curves for both polarities, they will see that both curves are similar and that they correspond to the same curve family.

5) If LE and RE are similar for both polarities, it is understandable that the most general characteristic, such as the diameter of the fibers, is not different for the polarity. Lines 327-329, “The average diameter of electrospun fibers, ??̅̅̅, in the random and the aligned fiber arrangements were 1.44 ± 0.22 and 1.54 ± 0.20 μm for PMMA +, and 1.54 ± 0.23 μm and 1.65 ± 0.15 μm for PMMA-, respectively . ”. I am agree with these results, but in addition, the authors need to perform a size distribution analysis taking into account the same number of intervals in each case and fitting their distributions to a Gaussian model.

6) It is generalized that the electrospinning technique favors the crystallinity of the polymer. In this way, the results of the loss of crystallinity during the process are striking. I suggest as possible controls to perform the DSC of 1) PMMA powder at 55 °C for 2.5 h, and 2) solvent-casting sample of PMMA dissolution. Although the crystallinity calculated from the calorimetric data is valid, if possible, carry out the determination by X-ray diffraction (XRD). The measurement of crystallinity by DSC is relative and depends on the area of ​​the melting peak that can vary depending on how the baseline for its integration is performed.

7) When analyzing density, the authors use as references the values ​​indicated in line 388-389: “Assuming that the theoretical density of the amorphous phase in PMMA is 1.17 g · cm-3 and the crystalline phase is 1.26 g · cm-3 [ 61.62], ". For a correct comparison, adjust the density values in Table 1 to two decimals.

8) The mechanical properties of PMMA fibers obtained by electrospinning with different polarity reveal changes in the microstructure of the fibers, this fact in itself is very interesting. I share with the authors that these changes are due to the different polymer-solvent distributions influenced by the polarity of the electric field. The conformation of PMMA in the fibers could be determined using FTIR microspectroscopy and X-ray microdiffraction. However, these changes in the charges distribution in the jet are not clearly demonstrated by the LE and RE dimensional parameters.

Author Response

Response to Reviewer 3:

Q1: Line 62-63. "In this study, we verify the effect of the positive or negative voltage on the mechanical performance of aligned and random poly (methyl methacrylate) (PMMA), fiber mats." Verification assumes prior knowledge; therefore, the authors are aware that the polarity of the electric field influences the mechanical properties of the PMMA fibers matrix. However, these antecedents are not indicated in the introduction, where only changes in the energy and potential of the surface of the fibers are indicated.

A1: The sentence was changed to the following, see page 2, line 62:

“In this study, we investigate the effect of the positive or negative voltage on the mechanical performance of aligned and random poly(methyl methacrylate) (PMMA), fiber mats.”

Q2: Line 82-83. “Several models describing the electrospinning process, and the behavior of the polymer jet were proposed; nonetheless, the effects of electrical polarity were neglected [24,27–32]. ”. In these models the polarity is not forgotten, it is implicit, and the models apply for both polarities. Operationally, during the electrospinning process, it is one of the first operational parameters to be set. In most cases, the polymer dissolution with a solvent or solvent mixture only produces fibers in a certain polarity. There are few exceptional cases, polymer-solvent(s) systems that produce fibers in both polarities.

A2: The sentence has been modified according to the Reviewer comment to the following, see page 2, line 81:

“Several models describing the electrospinning process, and the behavior of the polymer jet were proposed; nonetheless, the models did not consider the effect of the electrical polarity on the electrospinning process and the relation to the spinnability of polymer-solvent systems [24,27–32].”

Q3: In this sense, (comes from comment 2) the PMMA-DMF system seems suitable for studying the effect of the polarity of the electric field on the properties of fibers. However, the new correlation function ?̃? ~ ?̃^0.55 should be demonstrated for PMMA and other solvents and / or for different polymer-solvent(s) systems, before being proposed as a generalization.

A3: We agree that solvents as well as the humidity affect the electrospun polymer fibers and it may vary for different polymer-solvent systems. See the statement in line 133:

“Solvent evaporation and jet stretching and branching is known to cause a variation in the surface charge density at the jet surface [24,25,30,31,36].”

However, the approach we take in Eq. 2 considers no evaporation from the Taylor cone and the jet emerging from the tip of the cone.

Please follow also the text between line 25- 264, where the type of solvent is considered:

“The polymer jet length in the straight section is dependent on the polymer-solvent parameters, including viscoelasticity, electrical conductivity, dielectric constant, and surface tension [23,24]. By using DMF to dissolve PMMA, which has a relatively low vapor pressure, with a dielectric constant of (DMF) = 36.7 and electrical conductivity of (DMF) = 6×10-6 Sm-1 [39,40], electrical charge dissipation was rather low, allowing for storage of a higher amount of electrical energy in the liquid. The high dielectric constant of DMF encourages the ionization of species in solution, thus raising the electrical conductivity, and therefore the availability of charge resulting in larger volume charge density compared to other solvents under equal situations. Also, DMF often leads to liquid-liquid phase separation during electrospinning and to interior pore formation before fiber solidification [41].”

The electrical field strength E depends not only on the applied voltage but also on the polymer solution, where we take into account electrical conductivity and surface tension. Please follow the text in lines 270-275.

Q4: The new correlation function is based on the LE and RE values, taken from 10 samples. Lines 241-245: “For positive voltage (PMMA +), the length of the straight section of the jet after emerging from the needle (spinneret) was ?? = 2.9 ± 0.1 cm, and the radius of the disk of fibers deposited on the collector was ?? = 28.7 ± 6.3 cm. For negative voltage (PMMA-), the straight jet length was longer, reaching ?? = 3.2 ± 0.2 cm, while the disk of deposited fibers was smaller, with ?? = 24.2 ± 8.7 cm (Fig. 2 a-d).”. The values ​​between the two polarities appear not to be statistically different and correspond to the same distribution. In this sense, it is understood that the new correlation function is independent of polarity. Thus, in Figure 2, the authors show extreme cases of LE and RE, but they could also show similar cases. If the authors perform a statistical analysis of the LE vs E and RE vs E curves for both polarities, they will see that both curves are similar and that they correspond to the same curve family.

A4:  The differences in the polymer jet length are noticed, as the electric field strength varies for positive and negative electrical polarities in electrospinning. Thus, although the differences in the L_E are small, they are clearly indicated in Figure 2e, similarly to R_E extreme values. Noticeably, there is also the difference in the current values for both polarities, which indicates the changes in volume charge density and electrical stress responsible for polymer jet length. The higher accumulation of the charges affects not only L_E but also R_E. Therefore Figure 2 e-f) shows similar trends for both electrical polarities.

We have removed the following sentence related to the above comment:

Line 330: “. Interestingly, despite small differences in electrical current, significant differences in polymer jet length were visible.”

Additional clarification has been added, see line 331:

“Therefore, similar trends for both polarities are observed in the relation of the electric field to and .”

Q5: If LE and RE are similar for both polarities, it is understandable that the most general characteristic, such as the diameter of the fibers, is not different for the polarity. Lines 327-329, “The average diameter of electrospun fibers, ??̅̅̅, in the random and the aligned fiber arrangements were 1.44 ± 0.22 and 1.54 ± 0.20 μm for PMMA +, and 1.54 ± 0.23 μm and 1.65 ± 0.15 μm for PMMA-, respectively. ”. I am agree with these results, but in addition, the authors need to perform a size distribution analysis taking into account the same number of intervals in each case and fitting their distributions to a Gaussian model.

A6: The fiber diameter distribution, together with the Gaussian fitting, was added and presented in Figure 3S in the Supporting Information.

Q6: It is generalized that the electrospinning technique favors the crystallinity of the polymer. In this way, the results of the loss of crystallinity during the process are striking. I suggest as possible controls to perform the DSC of 1) PMMA powder at 55 °C for 2.5 h, and 2) solvent-casting sample of PMMA dissolution. Although the crystallinity calculated from the calorimetric data is valid, if possible, carry out the determination by X-ray diffraction (XRD). The measurement of crystallinity by DSC is relative and depends on the area of ​​the melting peak that can vary depending on how the baseline for its integration is performed.

A6: It is well known that the crystallinity of electrospun polymers differs from crystallinity of source materials formed by conventional techniques. There are two competitive mechanisms affecting final crystallinity of electrospun materials. The first one is related to the extremely fast solvent evaporation on the way of electrospinning resulting in crystallization hindering and formation of metastable low or even zero crystallinity structures. Such situation is quite common with lots of evidence in the literature [e.g. Enayati M.S., Behzad T., Sajkiewicz P., Bagheri R., Ghasemi; Mobarakeh L., Łojkowski W., Pahlevanneshan Z., Ahmadi M., Crystallinity study of electrospun poly (vinyl alcohol) nanofibers: effect of electrospinning, filler incorporation, and heat treatment, IRANIAN POLYMER JOURNAL, vol.25, No.7, pp.647-659, 2016]. The second mechanism acting in the opposite direction, i.e. leading to an increase of crystallinity, is related to the increase of molecular orientation on the way of jet formation which can stimulate crystallization leading to crystallinity increase. The first mechanism is much more common than the second one also supporting our DSC results. There is no doubt that the reduction of crystallinity for e-s fibers observed by us using DSC method is an experimental fact. The procedures for baseline corrections were applied according to all DSC rules providing reproducible and consistent results. The crystallinity values are presented in Table 2S in the Supporting Information.

Q7: When analyzing density, the authors use as references the values ​​indicated in line 388-389: “Assuming that the theoretical density of the amorphous phase in PMMA is 1.17 g · cm-3 and the crystalline phase is 1.26 g · cm-3 [61.62], ". For a correct comparison, adjust the density values in Table 1 to two decimals.

A7: The density values in Table 1 has been changed to two decimals, see Table 1, Line 414, Page 12.

Q8: The mechanical properties of PMMA fibers obtained by electrospinning with different polarity reveal changes in the microstructure of the fibers, this fact in itself is very interesting. I share with the authors that these changes are due to the different polymer-solvent distributions influenced by the polarity of the electric field. The conformation of PMMA in the fibers could be determined using FTIR microspectroscopy and X-ray microdiffraction. However, these changes in the charges distribution in the jet are not clearly demonstrated by the LE and RE dimensional parameters.

A8: As far as FTIR and X-ray experiments are concerned, we have made preliminary tests, but because PMMA is an amorphous polymer, it is not possible to obtain reliable results that would allow us to extend the discussion. Also, to verify that with FTIR we would need to run the test with PMMA fibers prepared with a different solvent. This is not the focus of this study.

See attached manuscript in PDF with tracked all changes.

Round 2

Reviewer 2 Report

If you deal with the length of jet in your future work, refer

to papers describing the influencing parameters in more detail.

Author Response

We thank the Reviewer for the comment and we will refer in further studies to the papers describing in detail the relevant parameters.

Reviewer 3 Report

1) It is crucial that the authors demonstrate that the LE vs. E plot with positive polarity is different from the LE vs. E plot with negative polarity (Figure 2e). The same for RE vs E plots (Figure 2f). r2 indicates the linear distribution of the data, but does not indicate that the curves are different.

2) The fiber diameter Size distribution and fitting distributions line in a Gaussian
model for PMMA samples (Figure 3S) shows that the analysis is not adequate because the fit to the Gaussian model is bad, the prediction of the diameter according to the models is low (see the calculated r2). It is necessary to recalculate the intervals in the distributions.

3) Determine the density of your PMMA powder sample and include this value in Table 1 and discussion in the text.

Author Response

Response to Reviewer 3:

Q1: It is crucial that the authors demonstrate that the LE vs. E plot with positive polarity is different from the LE vs. E plot with negative polarity (Figure 2e). The same for RE vs E plots (Figure 2f). r2 indicates the linear distribution of the data, but does not indicate that the curves are different.

A1: We tried to explain the observed phenomena in several places in the text:

P 3 Line 104-106

“In conclusion, although the effects of polarity reversal are expected to be small, they may exist for various reasons. Thus, for example, when the spinneret is negatively charged, the quantity of net charges of the flying jets could be smaller, resulting in a smaller electric current, as in fact, we have found.”

P 8 Line 308-309

“The electrical currents measured on the collector during electrospinning at positively and negatively voltage, were 90.3 ± 1.4 nA and - 86.8 ± 0.5 nA, respectively (Table 1S).

P 8 Line 319-322

”For example, increased volume charge density (e.g., PMMA+) should result in a greater initial acceleration of the polymer jet (per unit electrical field strength) against the decelerating forces (elastic tension, viscous stresses, and surface tension), resulting in a greater reduction in jet length, as was observed for PMMA+.”

We also tried to compare the results obtained for the positive and negative electrical polarity by fitting the experimental data with the following function y=ax+b^c., see the Supporting Information file, Figure 5S. New fitting clearly shows the differences between the two curves. Based on the above and the measured current it is expected that there is an increase of the jet length at different.

Regarding Re vs E we have stated in the manuscript in Page 7, line 283-284:

“The effect of the non-dimensionless electric field strength, , on the non-dimensional radius, , was negligible, although the dispersion of the data was rather large. “

Q2: The fiber diameter Size distribution and fitting distributions line in a Gaussian model for PMMA samples (Figure 3S) shows that the analysis is not adequate because the fit to the Gaussian model is bad, the prediction of the diameter according to the models is low (see the calculated r2). It is necessary to recalculate the intervals in the distributions.

A2: The fiber diameters were measured again on new samples we provide the average value with standard deviation, see page 9, line 339-341 and new Figure 1, page 10, line 362-363. Size distribution with Gaussian fitting of fiber diameters after new measurement was added in Figure 3S in the Supporting Information.

Q3: Determine the density of your PMMA powder sample and include this value in Table 1 and discussion in the text.

A3: Density of PMMA powder was added in the manuscript, see added text on page 12, Line 406:

“Assuming that the theoretical density of the amorphous phase in PMMA is 1.17 g·cm-3, which is similar for our PMMA powder ± 0.02 and the crystalline phase is 1.26 g·cm-3 [67,68],”

Please see attached the manuscript in PDF with marked all changes mentioned above.

Round 3

Reviewer 3 Report

Ok., I am agree with the answers.